# Transcatheter Aortic Valve Implantation in Younger Patients: A New Challenge

**DOI:** 10.3390/medicina57090883

**Published:** 2021-08-27

**Authors:** Giacomo Virgili, Salvatore Mario Romano, Renato Valenti, Angela Migliorini, Pierluigi Stefàno, Niccolò Marchionni, Nazario Carrabba

**Affiliations:** 1Cardiothoracovascular Department, Careggi Hospital, 50134 Florence, Italy; giacomo.virgili@unifi.it (G.V.); valentir@aou-careggi.toscana.it (R.V.); migliorinia@aou-careggi.toscana.it (A.M.); pierluigi.stefano@unifi.it (P.S.); niccolo.marchionni@unifi.it (N.M.); 2Experimental and Clinical Medicine, University of Florence, 50134 Florence, Italy; salvatore.romano@unifi.it

**Keywords:** aortic stenosis, TAVI, SAVR, low-risk patients, young patients, life-time management, bioprosthesis durability

## Abstract

The number of aortic stenosis patients in Western countries is increasing, along with better life conditions and expectancies. Presently, the volume of percutaneous transcatheter aortic valve implantations (TAVIs) is incessantly increasing, and has already overcome the surgical replacement procedure volume. According to the literature, TAVI is a feasible procedure even among low surgical risk patients, and American guidelines have extended the indications for TAVI, including shifting patient evaluations from high/low STS scores to old/young patients, a “paradigm shift” of aortic stenosis evaluation. As a result, low-risk young (<75 years-old) population management could be the next challenge in cardiology. To manage the life conditions of a 65 year old patient affected by aortic stenosis who is undergoing TAVI, one of the most crucial issue will be bioprosthesis durability and the appropriate intervention to make in cases of valve dysfunction or failure.

## 1. Introduction

Thanks to improved socioeconomic and health conditions leading to prolonged life expectancy in Western countries, aortic stenosis (AS) prevalence has remarkably increased [1]. According to a large meta-analysis and modelling study, the pooled prevalence of aortic stenosis in older (>75 years) populations of Europe, USA and Taiwan is 12.4%, and the prevalence of severe stenosis is 3.4% [2]. Though aging is exponentially linked with aortic stenosis prevalence [3], age-related degeneration is not the only pathogenic mechanism. In younger patients, aortic stenosis can be mostly due to the bicuspid aortic valve (BAV) or rheumatic disease. Replacement of the native valve is the treatment of choice, whereas a valve plasty is performed less frequently.

Over the past decade, more than 15,000 patients worldwide have been randomized in clinical trials concerning transcatheter aortic valve implantation (TAVI) procedure [4]. The volume of interventions is growing rapidly and, according to the Society of Thoracic Surgeons (STS) in the U.S., more than 300,000 TAVIs have been performed since the Food and Drug Administration’s first TAVI device approval in 2011 [5]. The STS-ACC TVT Registry (Society of Thoracic Surgeons–American College of Cardiology Transcatheter Valve Therapy Registry) reported in 2019 that the TAVI volume (*n* = 72,991) overcame all forms of surgical aortic valve replacement (*n* = 57,626), and today it is performed in all U.S. states [6]. In 2007, the first TAVI devices got a CE mark and entered into the European market, after which several first-in-men trails demonstrated their safety [7]. Presently, TAVI is the most frequent choice for treating aortic stenosis in older patients even in Europe. The German independent Institute for Applied Quality Improvement and Research in Health Care (AQUA) reported that the annual number of isolated surgical aortic valve replacement (SAVR) procedures decreased from 11,205 in 2008 (mean age, 69.8 years) to 9953 in 2014 (mean age, 68.5 years). Meanwhile, the volume of TAVI procedures has increased 20-fold from 2008 to 2014, and since 2013 has surpassed the annual numbers of isolated SAVR [8,9].

## 2. Guidelines

The first TAVI was performed by Alain Cribier in an inoperable patient in 2002 [10], and since that moment, transcatheter valve intervention has become an optimal alternative therapy to SAVR for patients with AS. TAVI was introduced in 2004 to treat comorbid patients at high surgical risk, avoiding cardiac arrest and cardiopulmonary bypass while reducing surgical trauma. During the subsequent years, modern transcatheter heart valves (THVs) have become more efficient, and the outcomes of TAVI have constantly improved. In 2017, European guidelines set out new recommendations for the assessment and treatment of AS patients [11]. Prospective randomized data obtained from the PARTNER B trial cohort demonstrated that TAVI is superior to medical therapy in inoperable patients up to five years after valve implantation [12,13,14]. Almost three randomized trials have compared the outcome of TAVI and SAVR in patients at high surgical risk [13,14,15]. Data up to five years have shown that TAVI is not inferior to SAVR, and that those patients who underwent transfemoral access obtained an additional advantage. Similar results have been obtained from randomized clinical trials of intermediate-risk patients, showing no difference in terms of one-year mortality between TAVI and SAVR, with the lowest mortality belonging to the transfemoral approach [16]. Furthermore, it has also been observed that, in TAVI patients, major bleeding complications were reduced and permanent pacemaker (PPM) implant occurred less frequently, while on the other hand data have showed higher risks of vascular access complications, paravalvular leakage (PVL) and atrial-ventricular block. TAVI studies have demonstrated that the percutaneous treatment of the aortic valve is a an optimal solution for bioprosthetic valve failure, as an alternative to re-do SAVR [17]. The valve-in-valve TAVI procedure has not been compared to SAVR in randomized clinical trials, but it has been demonstrated as very feasible, even in higher-risk patients. However, compared to SAVR, small-prostheses TAVI may implicate higher transprosthetic gradients and a need for PPM implantation [18]. In 2014, U.S. guidelines (ACC/AHA) approved TAVI for patients at prohibitive or high surgical risk, and in 2017 they extended TAVI indications to intermediate-risk patients. Finally, the latest (2020) U.S. guidelines have declared that even patients at low-risk of SAVR are eligible for TAVI under several circumstances [19].

The European Society (ESC) approved TAVI following a similar path to the U.S. guidelines. In 2012, TAVI was recommended for inoperable patients or patients considered high-risk. In the latest guidelines (2017), the TAVI procedure gained a class I level B recommendation for high-risk patients [11]. After PARTNER-3, Evolut low-risk and NOTION trial results [20,21,22], an enlargement of the population eligible for TAVI are expected to be included in the next European guidelines for valve heart disease; however, at present, TAVI remains a disservice to young patients, and clear evidence from randomized studies are necessary before TAVI can be extended to younger populations.

## 3. Younger Patients

Despite randomized clinical trials about TAVI showing encouraging results [12,13,14,15,16,17,18,20,21,22], it is important to notice that the mean age of the population was around 80 years and data about younger patients are lacking. Moreover, in these studies, patients with cardiac comorbidities, such as severe mitral and tricuspid valve disease and/or coronary artery disease, and anatomical relative contraindications, such as BAV, were excluded. The PARTNER 3 trial and the Evolut Low Risk Trial studied TAVI in low-surgical risk patients and again they showed the superiority of the balloon-expandable SAPIEN 3 valve and the non-inferiority of the self-expandable valve against SAVR, respectively [20,21]. Age is an important factor influencing the surgical risk, so it is possible to observe a reduction in age of studied population together with the decrease of the STS-score in a low-risk patient trial (the Evolut Low Risk Trial mean age is 74.0 ± 5.9 years). The cardiology community could expect that this trend is going to continue both in studies with a decrease of the mean age of the population and in clinical practice with even lower-surgical risk patients undergoing TAVI, as real life registries show [23,24,25]. We can consider ”young” as a 60 to 75 year old AS patient, and even if this population cohort has rarely been enrolled in TAVI clinical trials, they might ask for a percutaneous procedure or TAVI due to their clinical features (i.e., high surgical risk). It is possible to identify almost three kinds of young patient who can undergo TAVI instead of SAVR, and they are: asymptomatic degenerative AS patients, patients affected by bicuspid aortic valve with severe stenosis and young but high-risk or inoperable patients—the so-called “Cribier’s patient”.

Regarding asymptomatic patients, an update of the most recent European guidelines has been published [11], but even today this is a debated issue. However, data from the RECOVERY trial show a significantly lower incidence of operative mortality or cardiovascular death if an early SAVR procedure is performed in asymptomatic AS patients compared to medical therapy only; notably, the mean age of this population was 63.4 years [26]. 

Despite the lack of solid data about TAVI procedure in asymptomatic patients, in clinical practice, a certain number of these patients are treated percutaneously, but there are rational and scientific reasons guiding this choice. According to the literature, the presence of myocardial fibrosis in a hypertrophic left ventricle has been proven to be an independent predictor of all-cause mortality in AS patients [27]. Moreover, research is moving on, enough trials such as the EVOLVED trial (NCT03094143) and the EARLY-TAVR trial (NCT03042104) are studying asymptomatic patients. The former is currently enrolling asymptomatic severe AS patients with mid-wall late gadolinium enhancement to evaluate early SAVR outcomes such as all-cause mortality and AS-related hospitalizations [28]. The latter is randomizing asymptomatic patients comparing TAVI with clinical surveillance [29]. Even if further studies are needed, asymptomatic patients are likely to be a future cohort of population deserving TAVI.

A bicuspid aortic valve with severe stenosis is a unique situation with peculiar clinical characteristics, such as early age of presentation, and can be a technical challenge. In 2014, Mylotte et al. showed the feasibility of TAVI in BAV patients with good outcomes at short and intermediate term at the risk of a higher rate of aortic regurgitation after the procedure compared to tricuspid valves [30], other than higher bailout TAVI-in-TAVI and lower incidence of device success. The planning of the procedure is crucial, as is evaluating the patient’s clinical and anatomical features and choosing the right procedure for the right patient [31,32]. According to the large American TAVI registry, Makkar et al. showed similar 30 day (2.6 vs. 2.5%) and 1 year mortality (10.5 vs. 12.0%), as well as a similar 30 day and 1 year incidence of moderate-severe PVL, but an increased 30 day risk for stroke, comparing BAV with tricuspid aortic valve patients [33]. Recently, the BEAT registry compared BAV patients treated with Sapien 3 or with an Evolut R/PRO valve showing good procedural results with both bioprosthesis, and a higher rate of moderate-severe PVL at 1 year of follow-up in the Evolut R/PRO group and a more frequent annular rupture with Sapien 3 valves were observed [34]. So, mainly due to the ellipticity of the aortic annulus, the BAV stenosis treatment outcome is strongly influenced by bioprosthesis, and the literature data support self8-expandable valves [35].

Regarding “young” patients, we have to take into account that part of the population is at high surgical risk or is inoperable. These patients have a shorter life expectancy compared to the general population, and TAVI can be considered the only possible therapeutic option [34]. The OBSERVANT study observed 4801 patients younger than 80 years undergoing isolated TAVI or SAVR [36] with a logistic EuroSCORE significantly higher in TAVI patients across all age subgroups. TAVI patients under 65 years old showed the highest short- and long-term mortality as compared to older ones, mainly due to their clinical features instead of procedural complications [36]. Even in the YOUNG TAVR multicenter registry [37], analogous data can be observed where patients ≤75 years old presented more comorbidities, such as chronic obstructive pulmonary disease, diabetes and coronary artery disease, worse ejection fraction with a larger left ventricular end-diastolic diameter compared to older groups, but a lower STS-score. No difference in terms of all-cause mortality was observed at 30 days and 1 year, between the young and the older groups, but at 2 years of follow-up, the 76–86 year old group, namely the intermediate-age group, showed lower all-cause mortality. 

Therefore, “young” patients are a particular class who can benefit from TAVI, but with characteristic clinical features influencing their outcomes. Their STS score does not well describe the surgical risk they are facing, as age is an important item in the score calculation [37] and we should remember that everything begun with a young inoperable patient who became the first one treated by Cribier. 

## 4. The Paradigm Shift

Extending the TAVI procedure to low-risk patients is a new issue being debated in the cardiological community as, from this perspective, evaluating the suitability for TAVI is currently more driven by the age of the patient as opposed to their surgical risk. The need for a “paradigm shift” in patient evaluation is rising, with meticulous patient selection and tailored medicine becoming mandatory. 

In older patients, frailty is one of the most important issues to be taken into account. According to the literature, frailty is a “multidimensional syndrome characterized by decreased reserve and diminished resistance to stressors”, is both pathological or iatrogenic, is due to aging-related impairments [38], and is associated with prolonged hospital stay, complications, and all-cause short- and medium-term mortality [39]. Heart Team evaluation must take into account frailty when routinely evaluating patients, because in an extremely frail patient an interventional procedure may be completely futile, and even the latest European guidelines do not recommend intervention in these patients [11]. Frailty can be an important factor when weighing up the benefit/risk balance, but its assessment is still difficult because there is no validated or universally accepted scale. On the other hand, the STS-score may reveal inaccuracies in the patients’ risk assessment [40] and, according to the literature, is no longer a crucial discriminating factor. Hence, as the point of view of the cardiology community is changing, the methods adopted for patients’ evaluation should change too.

Some well-known TAVI issues become more important in younger patients, namely the durability of TAVI bioprosthesis by way of a longer life expectancy. A patient aged 65 years or less is considered “young” when at high/intermediate-surgical risk (and there are no doubts about the choice between TAVI and SAVR) but, more often, they are at low risk. Indeed, low-risk younger patients might be the larger, submerged part of the iceberg and, in the near future, a 60-year-old male patient might come into our office asking for TAVI, so we need clear evidence to answer this population’s queries. Cohorts of the PARTNER-3 and the Evolut low-risk trial [20,21] were around ten years younger than those in prior studies, with a mean age of 73 years and 74 years, respectively. Even in the NOTION trial the mean age was 79.1 ± 4.8 years [22]. Therefore, while today we are lacking solid data, we urge to get prepared for a probable near future.

## 5. The Durability Issue

Valve durability becomes a critical issue as a consequence of a better survival rate in low-risk compared to intermediate- and high-risk patients and of a prolonged life expectancy in younger individuals undergoing TAVI. No extensive data are yet available about TAVI bioprosthetic valves’ long-term durability, which also has been assessed with different methods and criteria. Gurvitch et al. [41] evaluated structural valve deterioration (SVD) and hemodynamic changes in 70 patients after TAVI with a balloon-expandable valve (median of 3.7 years), confirming a good medium- to long-term durability. The PARTNER 1 trial produced other long-term data on balloon-expandable valves and showed unchanged transvalvular gradient and aortic valve area over a 5-year follow-up [42]. Toggweiler et al. [43] showed a favourable outcome after TAVI in 88 patients, with signs of moderate prosthetic valve failure in 3.4% of patients and no cases of severe prosthetic regurgitation or stenosis at 5-year follow-up. Additionally, a good long-term performance at 5 years of transprosthetic gradient of self-expandable valves was observed in the Italian Clinical Service project [44], showing 1.4% of late significant prosthetic valve failure and asymptomatic degeneration with only mild stenosis in 2.8% of patients [44]. Recently, a longer follow-up for SVD and bioprosthesis valve failure (BVF) beyond 5-year follow-up has been conducted in eight studies (Table 1). Overall, at a follow-up of 5–8 years, moderate SVD was reported in 3.6–10.8%, severe SVD in 0–2.5%, and BVF in 0.6–7.5% of cases [45]. Following the recent European consensus [46], a satisfactory long-term performance of a self-expandable CoreValve was confirmed [47] by our group, showing a cumulative incidence function (CIF) of 2.7% of significant prosthetic valve failures and a CIF of 9.3% for moderate SVD, with none needing re-intervention. A lot of factors contribute to reducing the rate of SVD after the TAVI procedure, such as careful planning with precise measurement of the aortic annulus by a CT scan, the choice of the right prosthesis type and size for that specific patient, accurate deployment techniques, an adequate temporary pacing, and the perfect synchronization and communication among operators, assistants, and technicians of the team. Due to the interaction of the bioprosthesis with the native valve apparatus and the lack of suture-based anchoring, the risk of embolization and migration will remain part of the TAVI procedure, and it should be carefully monitored during the follow-up echocardiography [48]. Even for the occurrence of PVL after the TAVI procedure we could recommend the same behavior, though it should be strictly monitored [49,50]. Notably, more recent evidence obtained for the newer generations of CoreValve, Evolut R and Evolut Pro, showed a moderate–severe PVL rate of 3.4% and 0%, respectively [51,52]. Finally, the performance of percutaneous bioprosthetic valves appears to be reassuring and favorable even when compared with the outcome of surgical bioprostheses, showing >95% freedom from structural failure at 5 years [53], and from 60% to 90% freedom from valvular failure at 10 years [54,55]. However, larger dedicated studies are needed for determining the rate of structural deterioration of transcatheter valves over longer follow-up periods. With all this in mind, the implantation of TAVI in patients aged <75 years should currently be performed only under certain circumstances, such as in patients with comorbidities which severely increase their risk for SAVR, or in those included in controlled trials where outcomes are monitored.

## 6. Life-Time Management

It is conceivable that for the future of interventional cardiology, we can try to plan a sort of pathway for younger patients with AS. Charan Yerasi et al. [63] tried to imagine a near future with three different pathways, each with its own benefit/risk ratio. At present, the first approach for young patients is SAVR, according to the most recent guidelines and the lack of solid data. So, this approach leads to a SAVR-TAVI-TAVI strategy, with a surgical intervention at younger age. In this case, the dimension of the surgical bioprosthesis is of crucial importance, because it will likely receive two valve-in-valve TAVIs based on a 10-year predictable durability of surgical bioprostheses. No possibilities of a fourth percutaneous valve implantation can be imagined. Coronary access, after the first SAVR procedure, is feasible and TAVI-in-SAVR is not a new procedure. However, some risk of mismatch and coronary obstruction with the second (TAVI) and third (TAVI) intervention is not negligible. 

In patients undergoing TAVI who are at high risk for coronary obstruction, the intentional laceration technique of diseased bioprosthetic valve leaflets called BASILICA has been demonstrated to be effective and reasonably safe for preventing coronary artery obstruction. However, it is important to acknowledge that BASILICA might be not appropriate at low-volume centres in the real world and has not yet caught on in Europe [64]. Furthermore, achieving neo-commissural alignment during the initial TAVI has important clinical implications for future coronary reaccess and aortic valve reintervention, especially in younger patients, in which the lifetime treatment of aortic valves and coronary artery disease must be taken into consideration [65]. Recently, it has been suggested that a modified delivery system for the insertion technique during the initial valve deployment results in a better commissural alignment and less coronary overlap following self-expandable TAVI. However, this strategy should be confirmed in larger studies. Additionally, unanswered questions remain about the impact of commissural misalignment on balloon-expandable valve-in-valve TAVI, especially in patients with unfavourable aortic root anatomy [66].

Starting with a TAVI approach (TAVI-SAVR-TAVI strategy as a second pathway) lets us perform a subsequent SAVR in a 70 year old patient who still might tolerate cardiac surgery well, and finally a TAVI-in-SAVR approach would represent the third step at 80 years of age. TAVI in a young patient is an attractive option, with fast healing and no need for anticoagulation. In this way, SAVR as a second intervention is feasible, allowing the third and fourth TAVI procedures (TAVI-in-TAVI). However, one should realize that SAVR performed many years after TAVI is not as straightforward. Coronary access after the first TAVI cannot be easy, especially using self-expandable valves, even if technology may provide some solutions. Moreover, surgery for TAVI explantation may be challenging, especially in less experienced centres.

Finally, the third imaginable pathway is the “TAVI only” strategy (TAVI-TAVI-TAVI), so no surgical risk is present but it can be feasible only in a specific cohort of patients with particular anatomical features—the most important being a large aortic annulus.

## 7. Personalized Medicine and TAVI: The 3D Print Model

Individualized patient-center care is a central tenet of modern medicine. The variety of the transcatheter heart valves currently available affords the opportunity to select the most appropriate device for each individual patient. Prosthesis selection should be based on operator experience and, in the near future, on pre-procedural multimodal three-dimensional imaging.

Recently, 3-dimensional (3D) printing has emerged as a technique able to convert digital models into 3D objects [67], and cardiovascular models are particularly useful in interventional cardiology, where a deep knowledge of patient-specific anatomy is fundamental to guide catheter-based procedures. The valve annuls sizing is a cornerstone for the TAVI procedure. 3D printing allows to create patient-specific models of the aortic valve and aortic root anatomy, which is a revolutionary tool for planning TAVI. Moreover, 3D modeling may be useful for predicting the development of prosthetic regurgitation [68]. In particular, this method may help to simulate the implant and evaluate the best individual approach to TAVI. In addition, the tactile feedback provided by the 3D model may help in the Heart Team’s decision making process. Ensuring the reproducibility of a 3D model remains an open issue before employing the technique in clinical practice. The automatization of 3D model is desirable and great care when setting the threshold values and adjusting segmentation contours is necessary in order to avoid some errors that otherwise can be generated [69]. Even if the promises of this 3D printing model of the aortic root are great, only a limited experience has been done in the research area, and more data are needed for its validation [70,71].

## 8. Conclusions

After the first “Cribier’s revolutionary change” in the field of the treatment of aortic stenosis for high-risk or not operable patients, a second revolutionary change with a “paradigm shift” of the evaluation parameters is mandatory, aimed at assessing and balancing the efficacy and safety of the treatment of AS in younger patients. In this complex scenario, some patients will be more suitable for surgical and others for percutaneous AS treatment, reinforcing the concept of a personalized and tailored medicine. 

## Figures and Tables

**Table 1 medicina-57-00883-t001:** Long-term TAVI durability from the recent literature.

First Author	Bioprosthesis (*n*)	Years of Follow-Up	Results
Barbanti et al. [56]	CoreValve (*n* = 238) SAPIEN XT (*n* = 48)	8	BVF: 4.5% Moderate SVD: 5.8% Severe SVD: 2.3%
Holy et al. [57]	CoreValve (*n* = 152)	8	BVF: 4.5% Severe SVD: 0%
Carrabba N. et al. [47]	CoreValve/Evolut R (*n* = 182)	8	BVF: 0% to 4.5 Moderate SVD: 3.6–14.9% Severe SVD: 0–3.8%
Testa L. et al. [58]	CoreValve (*n* = 990)	8	Late BVF: 2.5% Moderate SVD: 3.0% Severe SVD: 1.6%
Eltchaninoff H. et al. [59]	Percut.ValveTech./Cribier-Edwards/SAPIEN/SAPIEN XT (*n* = 378)	8	BVF: 0.58% SVD: 3.2%
Antonazzo Panico R. et al. [60]	CoreValve (*n* = 278)	7	BVF: 2.5% Overall SVD: 3.6%
Murray et al. [61]	SAPIEN/CoreValve (*n* = 101)	7	BVF: 3.8% Moderate SVD: 8.9% Severe SVD: 1.3%
Deutsch et al. [62]	Corevalve/SAPIEN (*n* = 300)	7	Overall SVD: 14.9%

BVF = bioprosthesis valve failure; SVD = structural valve deterioration.

## Data Availability

Not applicable.

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
