# Peer review of "Transcatheter Aortic Valve Implantation in Younger Patients: A New Challenge"

_medicina, 2021, doi:10.3390/medicina57090883_

Round 1

Reviewer 1 Report

Quite an interesting review although with multiple questions marks.

As long as we understand its limitations, then TAVI has a role to play.

TAVI is not the “magic bullet” but unfortunately we live in a market-driven society which does not help at all. Once again, we are facing another unnecessary debate similar to the boring issue between PCI versus CABG and off-pump CABG vs on-pump CABG.

We have more options available but it does not mean that they must be used regardless.

At present, TAVI remains a disservice to young patients. The original indication should remain the norm.

The review is quite comprehensive although it should be re-addressed with a more impartial attitude. A patient-specific approach has been briefly mentioned although it would be worth expanding a little bit more on the subject.

The English language should be reviewed .

Author Response

First of all, many thanks for your comment. We agree with you, another debate about the issue between TAVI vs. SAVR in young patients is unnecessary. According to your suggestion, we added these new sentences:”However, at present, TAVI remains a disservice to young patients, and clear evidences from randomized studies are mandatory before the extension of TAVI indication to younger population”, as you can see in the revised manuscript, paragraph 2, page 2, line 88.     

Moreover, we added:” Individualized patient-center care is a central tenet of the modern medicine. The variety of transcatheter heart valves currently available affords the opportunity to select the most appropriate device for each individual patient. Prosthesis selection should be based on operator experience and, in a near future, on pre-procedural multimodal three-dimensional imaging”, as you can see on the paragraph 7, page 7, the first line (306) of the revised manuscript.

Eventually, we tried to improve the English language and to correct any typos errors through the manuscript, mostly depending on the format (automatic syllabication) provided by the platform of uploading.

With my personal best regards,
Nazario Carrabba, MD.

Reviewer 2 Report

A very well written editorial. Amazing work. Please recheck the manuscript as there were minor spelling errors.

As one describes the 3 ways of TAVI and SAVR pattern in young it should be acknowledged that surgery many years after TAVI is not as straightforward.

Author Response

First of all, we are obliged to you for your comment. According to your suggestions we added this new sentence:” However, one should realize that SAVR performed many years after TAVI is not as straightforward”, as you can see on the revised manuscript, paragraph 6, page 7, line 295.   

Moreover, we tried to improve the English language and to correct any typos errors through the manuscript, mostly depending on the format (automatic syllabication) provided by the platform of uploading.

With my personal best regards,
Nazario Carrabba, MD.

Reviewer 3 Report

I want to congratulate authors for this editorial. It is well structured and the revision is appropriate and well performed. Otherwise, I think 2 aspects must be consider:

  • I missed in the life-time management part longer discussion about important issues like re-access to coronary arteries, reviewing evidence about BASILICA and TAVR-coronary alignment.
  • English language. There are some mistakes of language that must be corrected to accept this paper.
    • a. In the abstract: in line 11 it is written country and must be said: countries. 
    • b. In same line, implatantion>>>> it is implantation
    • c. Line 14 guide-lines should be changed for guidelines.
    • d. Line 139 it is written sim-ilar and this word cut is not correct.
    • e. In line 71 para-valvular leak abbreviation is properly written as PVL. In line 234 it is repeated and it is not necessary.
    • f. In line 225: It is said: CIF (cumulative incidence function) and the correct form to describe term and abbreviation is other way around: Cumulative incidence function (CIF).
    • Please, take care of language, it is important to consider your submission.

Author Response

Primarily, we are grateful for your encouraging words. According to your suggestion, we added new sentences:” In patients undergoing TAVI at high risk for coronary obstruction, the intentional laceration technique of diseased bioprosthetic valve leaflets called BASILICA has been demonstrated to be effective and reasonably safe for preventing coronary artery obstruction. However, it is important to acknowledge that BASILICA might be not appropriate at low-volume centres in real-world and has not yet been catching on in EUROPE (Lederman RJ, et al. Preventing Coronary Obstruction During Transcatheter Aortic Valve Replacement: From Computed Tomography to BASILICA. JACC Cardiovasc Interv. 2019 Jul 8;12(13):1197-1216. doi: 10.1016/j.jcin.2019.04.052). Furthermore, achieving neo-commissural alignment during initial TAVI has important clinical implications for future coronary reaccess and aortic valve reintervention, especially in younger patients, in which the lifetime treatment of aortic valve and coronary artery disease must be taken into consideration (Tang GHL, et al. Alignment of Transcatheter Aortic-Valve Neo-Commissures (ALIGN TAVR): Impact on Final Valve Orientation and Coronary Artery Overlap. JACC Cardiovasc Interv. 2020 May 11;13(9):1030-1042. doi: 10.1016/j.jcin.2020.02.005). Recently, has been suggested that a modified delivery system insertion technique during initial valve deployment results in better commissural alignment and less coronary overlap following self-expandable TAVI. However, this strategy should be confirmed in larger studies. Additionally, unanswered questions remain about the impact of commissural misalignment on balloon-expandable valve-in-valve TAVI, especially in patients with unfavourable aortic root anatomy (Tang GHL, et al. Feasibility of Repeat TAVR After SAPIEN 3 TAVR: A Novel Classification Scheme and Pilot Angiographic Study. JACC Cardiovasc Interv. 2019 Jul 8;12(13):1290-1292. doi: 10.1016/j.jcin.2019.02.020)“ as you can see on the revised manuscript, paragraph 6, page 6, line 275.
Furthermore, we reported the new references number 64, 65 and 66 on the revised manuscript.

In the end, we tried to improve the English language and to correct any typos errors through the manuscript, mostly depending on the format (automatic syllabication) provided by the platform of uploading. According to your suggestion, every description of term and abbreviation has been revised and edited.

With my personal best regards,
Nazario Carrabba, MD.

Round 2

Reviewer 1 Report

The authors have addressed my comments and suggestions.